# Reuse of Buffing Dust-Laden Tanning Waste Hybridized with Poly- Styrene for Fabrication of Thermal Insulation Materials

**Wajad Ulfat** [1]**, Ayesha Mohyuddin** [1,]*[ ], **Muhammad Amjad** [1]**, Tonni Agustiono Kurniawan** [2,]*[ ],
**Beenish Mujahid** [3]**, Sohail Nadeem** [1]**, Mohsin Javed** [1]**, Adnan Amjad** [1]**, Abdul Qayyum Ashraf** [4],
**Mohd Hafiz Dzarfan Othman** [5]**, Sadaful Hassan** [1] **and Muhammad Arif** [1]

1   Department of Chemistry, School of Science, University of Management and Technology,
    Lahore 54770, Pakistan
2   College of Ecology and the Environment, Xiamen University, Xiamen 361102, China
3   Department of Architecture, School of Architecture and Planning, University of Management and Technology,
    Lahore 54770, Pakistan
4   Department of Biotechnology, Faculty of Science and Technology, Virtual University of Pakistan,
    Lahore 51000, Pakistan
5   Advanced Membrane Technology Research Centre (AMTEC), Faculty of Chemical and Energy Engineering,
    Universiti Teknologi Malaysia (UTM), Skudai 81310, Johor Bahru, Malaysia
*   Correspondence: ayesha.mohyuddin@umt.edu.pk (A.M.); tonni@xmu.edu.cn (T.A.K.)

**Abstract:** Air pollution, resulting from buffing dust waste produced by local leather tanning industry, has become a critical issue for the environment and public health. To promote a circular economy through resource recovery, this work developed a thermal insulation composite using buffing dust-laden tanning waste mixed with polystyrene and a blowing agent. To prepare the samples from leather tanning waste, different proportions of buffing dust (5–20% ($w/w$)) were blended with polystyrene in the presence of 3% ($w/w$) blowing agent. The composite material was processed in double-barreled with co-twin extruder to expose it to pressure and then heated at 200 °C. Different physico-chemical properties of composite samples were determined. The prepared composite materials had a good thermal conductivity (0.033–0.029 W/m-K), strong compression (5.21–6.25 ton), density (38–20 kg/m$^3$), and water absorption (5–7.5%), as compared to conventional constructional insulation panels. The thermal conductivity of polystyrene was reduced to 10% after the addition of buffing dust (20% $w/w$). The presence of a blowing agent in the composite material enhanced its volume without compromising its physico-chemical properties. Thermo-gravimetric analysis showed that the thermal stability of the composite material ranged from 200–412 °C. FTIR analysis indicated that the composite had carbonyl and amino functional groups. The SEM images revealed the formation of voids with a decreasing homogeneity of the composite after the addition of the buffing dust waste. The EDX analysis revealed that the composite also had 62% of C and a tiny amount of Cr. This implies that the composite panels can be used for installation in buildings as thermal insulators in the construction sector. Overall, this work not only resolved the energy consumption problems during manufacturing, but it also brought positive impacts on the environment by recycling hazardous buffing dust and then reusing it as a thermal insulation material. Not only does this reduce the air pollution that results from the buffing dust waste, but this also promotes resource recovery in the framework of a circular economy.

**Keywords:** buffing dust; circular economy; construction; polystyrene; resource recovery; waste





## 1. Introduction

As the second largest industry in Pakistan, the tanning industry grows day after day due to economic growth and industrialization. About 800 units of tanneries are located in Kasur city (Pakistan). They produce a large amount of buffing dust-laden tanning waste that contributes to air pollution in the city. As a result, recently, air pollution has increased

to an alarming point due to the waste [1]. As a hazardous waste, the buffing dust contains Cr(III). Under certain conditions, it is converted into Cr(VI). Traditional waste disposal methods such as landfills and incineration could not remove the hazardous nature of the toxic metal to the environment [2–4]. Consequently, the tanning waste causes adverse effects on public health [5,6].

These effects have worsened due to toxic Cr(III)-contained sludge produced during leathery processing. During the processing, only 20% of the total raw material is obtained as a final product [7,8], while buffing dust and chrome shavings are produced with 35–40% ($w/w$) in tanneries along with other residues [6,7]. As the solid waste in tanneries is commonly disposed through incineration, the emission of $NO_x$ and $SO_x$ during the disposal contributes significantly to air pollution [5]. Furthermore, the leaching of harmful chemicals during landfilling leads to water contamination and soil pollution [7,8]. As a result, scientists have worked to reduce the amount of the industrial waste, while exploring ways for recycling and reusing it after incorporating technological values into the waste. For the sake of sustainability, the waste needs to be reduced, recovered, recycled, and reused in such a way that it could be turned into a valuable resource such as a thermal insulation material to promote resource recovery and a circular economy. Resource recovery minimizes waste generation by utilizing by-products that remain from other processes, thus adopting a zero-waste approach.

As compared to linear economy, which results in environmental challenges such as $CO_2$ emission, a circular economy facilitates waste prevention/reduction and its recovery and recycling to deal with the impacts of waste disposal on the nexus of society, environment, and economy. As an economic system based on the reuse and regeneration of materials or products, this economic model serves as a means of continuing production in a sustainable or environmentally friendly way by reverting excessive rates of waste generation and resource consumption. In contrast, failing linear economic models treat resources as if they are infinite. Unlike a circular economy, the linear model is unsustainable.

For this reason, in recent years, researchers have intensified the search for diverse applications of unused buffing dust for thermal insulation materials in its composite form [9–11]. Currently, different types of boards, prepared from waste materials, have been found to be compatible with physico-mechanical standards. Specifically, the recycling of *Balsa* wood waste, jute, and sugarcane bagasse into strand boards with multi-layer particles in the form of polymer composites has motivated the researchers to uncover the possibilities of tanning waste for recycling purposes [4,10].

Just recently, scientists have explored the recycling of unused waste obtained from tanneries. The reuse of tanning waste as rubber fillers has been one of the most promising applications [10–13]. The leather waste, such as chrome shavings and buffing dust, has incorporated different types of rubber such as carboxylated butadiene nitrile and styrene butadiene to enhance the strength of bonding with elastomers and fillers [14,15]. The potential of $CH_4$ production from tanning waste was economically attractive in anaerobic thermophilic digester process [16]. Buffing dust consists of proteins, and it can be utilized for $CH_4$ production through anaerobic digestion [16–18]. However, the existence of cationic polyacrylamide flocculant, widely used for activated sludge dewatering, slows down $CH_4$ production in anaerobic digesters [19–21].

Similarly, buffing dust with a different quantity of oxygen during incineration resulted in bottom ash, and was further used as a stabilizer in Portland cement. Furthermore, its metal capacity was completely adsorbed into cement block with specific characteristics [22]. Unused waste such as chrome shavings and buffing dust has low thermal conductivity. The waste could be utilized to prepare thermal insulation panels, bricks, plaster, and cement blocks [22–28]. Buffing dust could form 5% of the filler in bricks [29–33].

The International Energy Agency (IEA) reported that residential and commercial buildings consumed 15,082–16,843 trillion BTUs of energy [34]. Similarly, the US Energy Information Administration reports that residential buildings consumed 52% of energy for heating process [35]. Hence, this energy consumption can be minimized through technical

ways such as thermal insulation of buildings [36]. Commonly available ceiling panels are costly and made from different insulation materials such as polyurethane, fiberglass, wood, polystyrene, hemp, perlite, expanded clay, vermiculite, cotton, and plaster of Paris [37]. Ordinary thermal insulation materials have thermal conductivity ranging from 0.034 to 0.173 W/m-K [38,39]. Suitable low cost materials could be used to enhance the efficiency of panels and reduce their thermal conductivity.

Environmentally friendly and energy saving properties of buffing dust waste into useful material production has been explored in recent years. For example, conventional acoustical ceiling panels (ACT) have been tested as insulation materials. However, their utilization could not stop heat exchange effectively in buildings [40]. Despite a variety of works have incorporated buffing dust in construction materials for recycling and reuse purposes [41], none has uncovered its novelty in the form of composite as thermal insulation materials. The waste in its composite form is stable to stop heat transfers in buildings. This material is also biologically stable, resulting from the complexation between collagen carboxyl groups and salts of Cr(III) [42,43].

To the best of our knowledge, buffing dust has not been mixed with polystyrene to form composites in the presence of a blowing agent. To bridge knowledge gaps in the body of literature, this work presents the technical feasibility of recovering leather tanning waste for producing insulating materials from buffing dust-laden tanning waste. Al-though it is widely known that the recovery of tanning waste already took place, this work utilized residual polishing powder as starting materials with compatible physico-chemical and mechanical properties.

To reflect its novelty, the work prepared and fabricated thermal insulation materials obtained from polystyrene and buffing dust in the presence of a blowing agent. To enable the composite to have low thermal conductivity and to increase compressive strength, the buffing dust was mixed with polystyrene. Installing it in buildings as thermal insulators is important because the materials could reduce energy consumption, while allowing the recovery of tanning waste and the recycling of polishing powder to cope with hazardous air pollution, resulting from the tanning industry.

## 2. Materials and Methods

### 2.1. Materials

Polystyrene $(C_8H_8)_n$ was collected from Diamond Jumbolon factory in Lahore (Pakistan). Density, molecular weight, and thermal conductivity of the polymer were 600 kg/m$^3$, 15,200 g/mol, and 0.038–0.040 W/m-K, respectively. This polystyrene was used to prepare the thermal insulation panels as expanded polystyrene (EPS). Buffing dust waste was collected from a leather industry in Kasur (Pakistan). The tanning waste was produced during the finishing process of the leather. The cowhides were used as raw materials to produce leather in the tannery. Buffing dust was dried under sunlight and stored in plastic sacks before using it at room temperature. The moisture value of this sample content was 15%.

As a blowing agent, azodicarbonamide (ADC) was obtained from Henan Xingyang Chemical Industries (China). Its appearance was light yellow in powder form. It took processes for synthetic materials, widely used in the pressurized and non-pressurized blowing of polyethylene, polyvinyl chloride, natural and synthetic rubber. Ordinary ADC decomposed at 200–220 °C to meet the requirements of plastics and rubber processing.

### 2.2. Methods

Composite materials were synthesized by copolymerization of polystyrene with varying concentrations of buffing dust from 5 to 20% ($w/w$). They were labeled as 5-BD to 20-BD, respectively, with 3% ($w/w$) of ADC as a blowing agent. All the materials were co-polymerized in a cotwin-screw extruder with barrel being shaped with heating apparatus. All the materials were copolymerized at 210 °C at a suitable pressure. The molten copolymer product was synthesized in an extruder and then poured it into steel molds.

The product was allowed to cool at ambient temperature for 30 min. Then, molds were opened, and composites were stored for analysis (Figure 1). The dimension of the samples was 4 inch in length, 2 inch in width, and 1 inch in thickness that were widely used for the physio-mechanical tests (water absorption, compression strength, density). A half-inch thick circular sample plate was used to determine thermal conductivity with a 2.5-inch diameter.

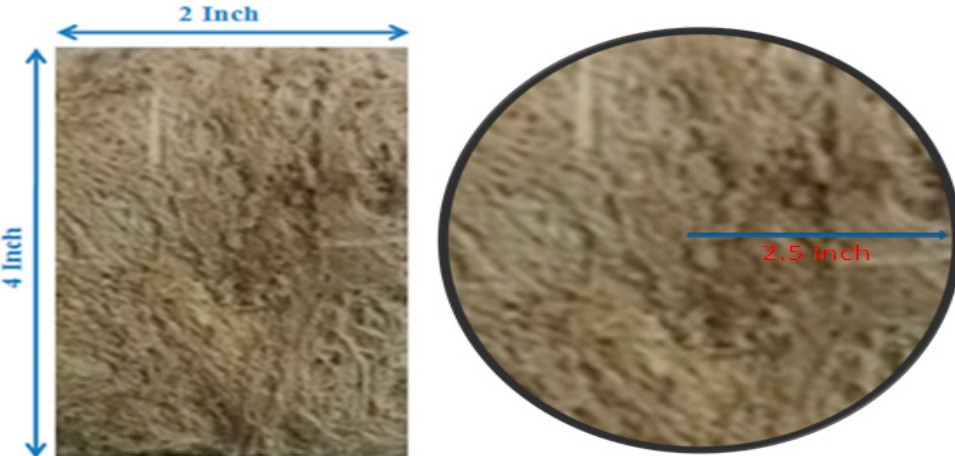

**Figure 1.** Representative samples for analysis.

### 2.3. Characterization

The physico-chemical and mechanical properties of the prepared samples (compression strength, density, and water absorption) were determined based on the insulation materials. Lee's Disc apparatus was used to determine the thermal conductivity of composite samples at ambient temperature [44]. The apparatus of thermogravimetric analysis (model TGA-50 SrC30025100553, New Orleans, LA, USA) was used to measure the thermal stability of the samples. Fourier transform infrared spectroscopy (FT-IR) model Agilent Cary II-630 (US) was used to analyze the samples with KBr running at 650–4000 cm$^{-1}$ range. Scanning electron microscope (SEM) model (FEI Nova Nano SEM 450, Sydney, Australia) was used to determine the morphology and homogeneity of the composite, while energy dispersive X-ray (EDX) spectroscopy model (Oxford INCA x-actEDX, US) was used to determine their chemical elements [45].

## 3. Results and Discussion

### 3.1. Physico-Chemical and Mechanical Properties

The test results of compression strength, density, water absorption, and thermal conductivity of the prepared samples are presented in Table 1. The compression strength of composite samples improved with the increasing quantity of buffing dust [46] (Figure 2a). The compression strength of pure polystyrene was 5.21 tons. After adding 20% (*w/w*) buffing dust, its compression strength increased by 20% to 6.25 ton. Hittini et al. [47] reported that the compression strength decreased by 25% from 15.32 MPa to 11.55 MPa due to the addition of buffing dust with polystyrene and chemical processing. The differences in compression strength were due to the blowing agent and chemical treatment used in this work.

Composite material's density decreased after the addition of buffing dust (Figure 2b). Around 50% of the decrease in density was observed when 20% (*w/w*) of buffing dust and the blowing agent was mixed with polystyrene. A similar trend of density was reported by Lakrafli et al. [8], who found that the density decreased by 42% from 2.24 to 1.31 g/cm$^3$ after the addition of buffing dust in cement and reduced by 23% from 1.57 to 1.21 g/cm$^3$ in the case of plaster of Paris and buffing dust composite. The water absorption of the samples was enhanced with the increasing value of buffing dust (Figure 2c) due to the creation of voids in the composite after the addition of a blowing agent. The thermal conductivity

of material with 20% (*w/w*) buffing dust (0.029 W/m-K) was 12% lower than the ceiling panels of pure polystyrene (0.033 W/m-K). The decreasing trend of thermal conductivity is represented in Figure 2D.

**Table 1.** Physico-chemical properties of samples.

| Sample | Compression Strength (ton) | Density (kg/m$^3$) | Water Absorption (%) | Thermal Conductivity (W/m-K) |
|---|---|---|---|---|
| Pure polystyrene | 5.21 | 38 | 5 | 0.033 |
| 5-BD | 5.61 | 33 | 5.6 | 0.032 |
| 10-BD | 5.80 | 29 | 6.3 | 0.031 |
| 15-BD | 5.95 | 25 | 6.9 | 0.030 |
| 20-BD | 6.25 | 20 | 7.5 | 0.029 |

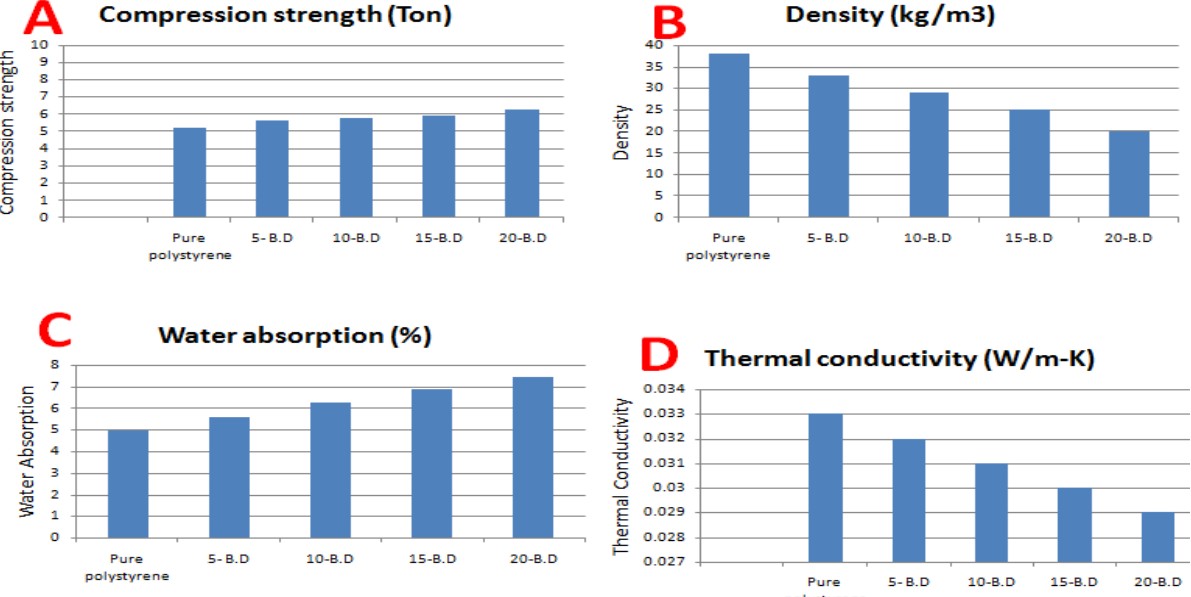

**Figure 2.** Physico-chemical properties of samples. *Remarks:* (**A**). Compression strength; (**B**). Density; (**C**). Water Absorption; (**D**). Thermal conductivity.

Hittini et al. [47] also reported the same trend: that the thermal conductivity of composite from pure polystyrene (0.0515 W/m-K) and 10% buffing dust was further lowered to 0.0447 W/m-K. It was about 13% lower than the pure polystyrene. By adding the blowing agent in the composite material, its volume was enhanced to 40–50%. This decreased its density without compromising material's properties. The decreased density helped the installation of the composite panels in the construction work. The blowing agent had no influence on the thermal conductivity of the composite.

Analysis of physico-chemical properties of composite samples with different ratios of buffing dust and polystyrene implied that 20% (*w/w*) buffing dust composite panels was the best for insulation purpose. Hence, only panels with 20% (*w/w*) buffing dust composite were characterized using TGA, FTIR, SEM, and EDX.

### 3.2. Thermogravimetric Analysis (TGA)

The stability of composite materials on heating was calculated based on the TGA analysis. The TGA graph represents the trend of weight loss of the material (Figure 3). The composite was heated over 600 °C. The composite material was thermally stable at 200 °C, as reflected by the TGA line. As the degradation of the material was observed from 200–280 °C, the composite rapidly degraded after 320 °C, while 50% of the composite's weight was lost at 385 °C. Almost the total weight of material was degraded at 412 °C.

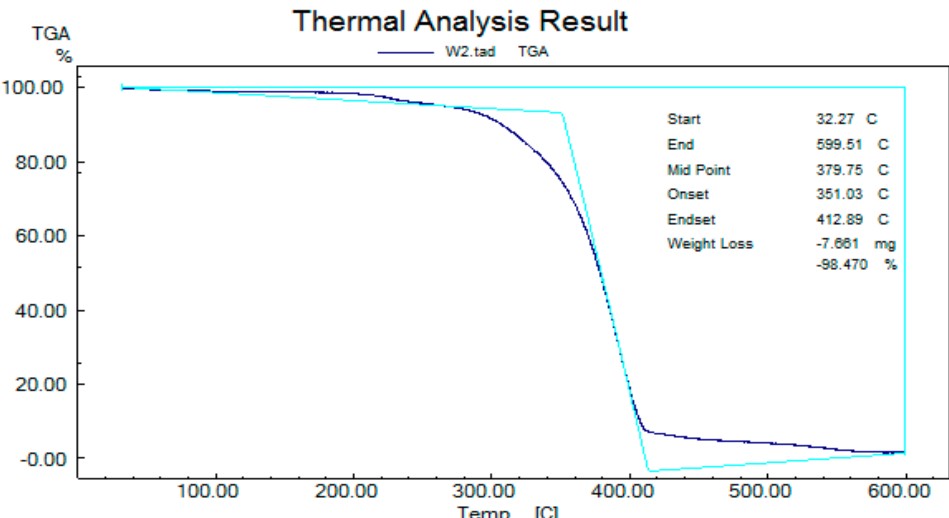

**Figure 3.** Thermo−gravimetric Analysis (TGA).

In their previous studies, Hittini [47] reported that the composite material was stable at 280–402 °C, while this composite material was thermally stable at 200–412 °C. Thermal stability is important for insulation materials to prevent it from heating shrinkage. As the composite had a good thermal stability, the material can be used in insulation panels as an alternative in the construction industry.

### 3.3. FT-IR Analysis

The FT-IR spectra of buffing dust and composite material are presented in Figures 4 and 5, respectively. For the buffing dust spectra, the band at 3293 cm$^{-1}$ represented the H- bonding that was associated with -OH groups. The vibration of the N−H stretching was represented by the band at 2928 cm$^{-1}$, while its alkyl groups had the C−H stretching at 2167 cm$^{-1}$. The band at 1625 cm$^{-1}$ represented the C=O group in the buffing dust. The bands between the 1400–1600 cm$^{-1}$ represented the –NH groups in the buffing dust.

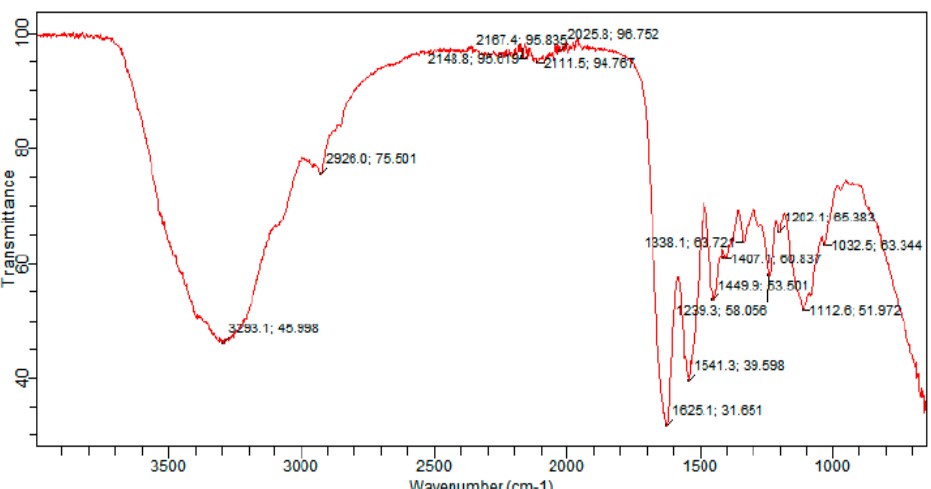

**Figure 4.** FT-IR spectra of buffing dust waste.

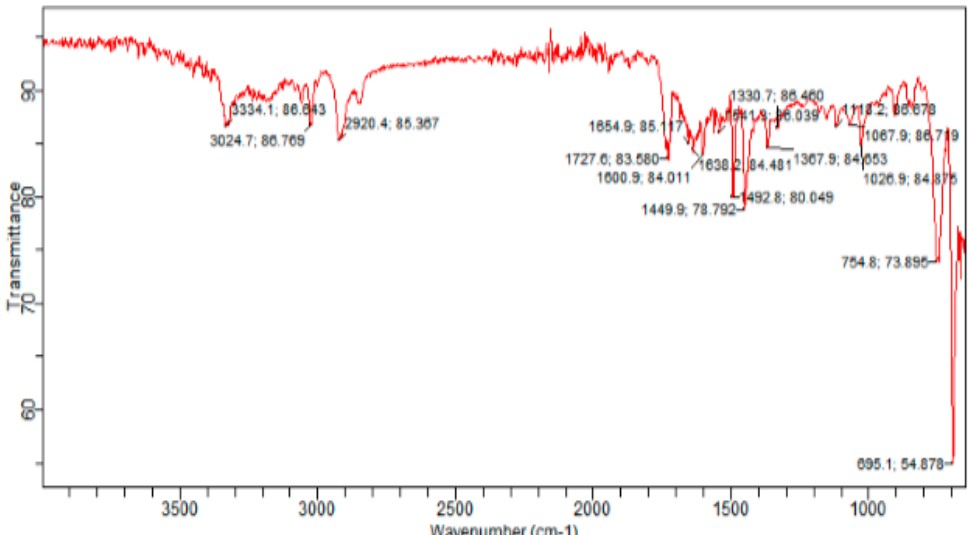

**Figure 5.** FT-IR spectra of composite sample.

In their previous studies, Basak et al. [48] also reported the FT-IR analysis of the buffing dust at 1656 and 3077 cm$^{-1}$, similar to those of this study. They found the bands to be between 3400–2900 cm$^{-1}$, which implied that the -OH groups were attached by H-bonding [48]. In the literature [24], the FT-IR spectra of pure polystyrene existed at the following bands at 3025, 2920, 2846, 1696, 1598, 1491, 1452, 1448, 1031 cm$^{-1}$. In this work, the FT-IR spectra of the composite existed at 3334 and 3024 cm$^{-1}$ that represented the H- bonding of –OH groups and N−H stretching, respectively. The bands at 2920 and 1727 cm$^{-1}$ were attributed to the N−H stretching and the C=O groups. The results suggested that the H-bonding was present between the polystyrene and buffing dust.

### 3.4. SEM Analysis

The morphology of the composite was examined by using SEM. The SEM analysis of this material, presented in Figure 6, showed voids created in the insulation material. This was attributed to the presence of the blowing agent. The homogeneity of the composite decreased due to the mixture of blowing agent and buffing dust. Buffing dust and blowing agent particles were embedded on the polystyrene polymer chain (Figure 6d). Some particles showed irregularity in the composite (Figure 6e,f). The buffing dust in the composite not only reduced the thermal conductivity, but also enhanced its compression strength. Abu-Jdayil et al. [13] also reported similar findings of polystyrene composite to those of this work.

### 3.5. EDX Analysis

The type of the elements in the composite is presented based on the EDX analysis of the sample (Figure 7). The sample had 62% of C because polystyrene and buffing dust were organic materials. Chromium (0.2%) was also observed because the transition metal was involved in the tanning process of the leather in the form of [Cr(OH)SO$_4$]. About 4% of O was present in the sample due to the fact that Cr was attached with the collagen of the gelatin of the sample. The collagen had active sites on carboxylate groups, where the metal was bonded with it in a complex form. If the Cr(III) is oxidized and converted into Cr(VI), the metal becomes carcinogenic. S and Cl were also present in the composite due to the involvement of the chromium salts in the leather processing [49,50].

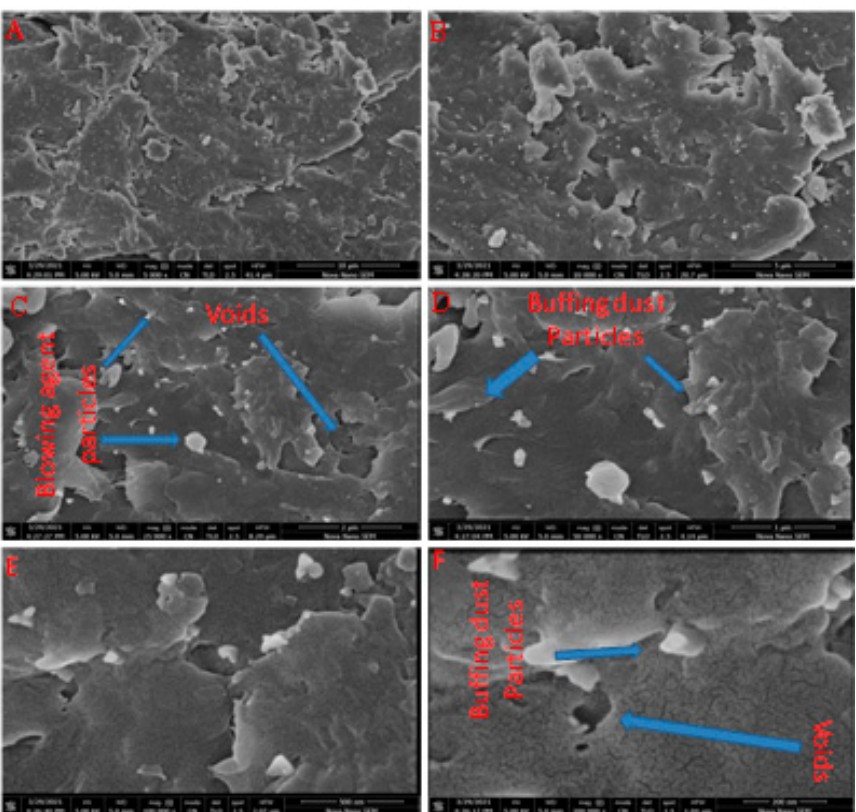

**Figure 6.** SEM analysis of composites with different resolutions. (**A**): 10 nm; (**B**): 20 nm; (**C**): 40 nm; (**D**): 80 nm; (**E**): 100 nm; (**F**): 150 nm.

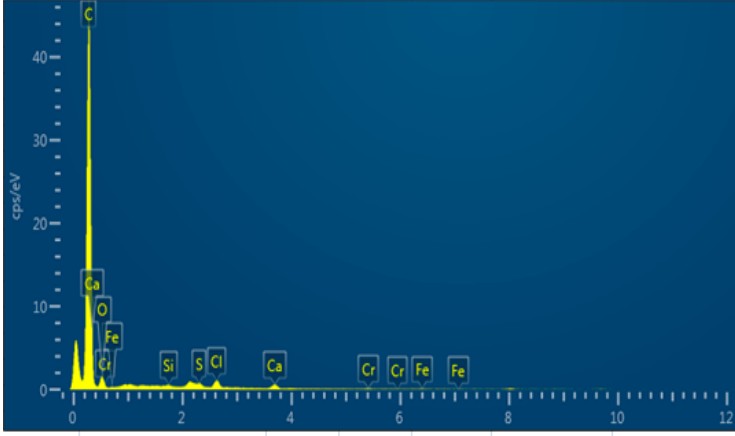

**Figure 7.** Energy dispersive X-ray analysis of composite.

## 4. Implications of This Work on Circular Economy

To adopt a zero-waste approach in promoting sustainability, for the sake of waste avoidance, industrial users strive for material recovery from downstream processing to address air pollution problem due to buffing dust from tanning industry [51]. This work has uncovered practical applications for synthesizing thermal insulation panels that can be used for the ceiling of buildings in construction industry [52].

The recovery, recycle, reuse, conversion, and re-introduction of the buffing dust waste into a supply chain of ceiling panels for the production of insulation materials contribute to a sustainable production and consumption in the construction industry with respect to circular economy for cost-saving/cost-recovery opportunity [53]. If the growth of the buffing dust waste could be reversed by tackling it from upstream to downstream, this

facilitates a sustainable resource recovery of by-products in the tanning industry [54]. With respect to the logistics of such implementations, it is not economically feasible in areas with a minimum leather tanning industry. To reverse the growth of the buffing dust waste from upstream to downstream, an extensive presence of a leather tanning industry in the same areas is necessary to produce ceiling panels.

This work represents a practical solution for local air pollution problems due to buffing dust waste because the waste can be recycled and reused to produce materials with added value after being mixed with polystyrene to improve its compressive strength [55]. This represents a closed-loop of circular economy that ultimately benefits the construction industry [56]. Shifting from a linear economy to a circular economy is vital to manage buffing dust waste efficiently and effectively [57].

The insulation panels bring socio-economic benefits that can minimize energy consumption [52]. This reduces the tanning waste disposed of in local landfills [58]. This paradigm enables environmental problems due to waste generation to be tackled locally to bring tangible impacts on the environment in the long-term. In the future, additive materials and new methods can be tested to enhance the physico-chemical and mechanical properties of the buffing dust waste such as water absorption, density, mechanical strength, and thermal conductivity.

## 5. Conclusions

Buffing dust is a tanning waste produced abundantly by local leather industry. This work has demonstrated that incorporating buffing dust with polystyrene in the composite has improved its thermal insulation. The results implied that water absorption and compression strength of the sample material have increased. However, their density and thermal conductivity decreased after the addition of buffing dust in the composite. The density of the material was compatible with thermal insulation panel (30–80 kg/m$^3$, EN 253). The composite material had a superior compression strength (598.5 kPa), higher than the minimum requirement (70–250 kPa, BS EN 826) for insulation panels.

The increasing water absorption property of the composite material was 1.07%, close to the acceptable limit of 1% (ASTM C553). After the addition of buffing dust, the thermal conductivity of the sample decreased, better than the minimum requirement [0.024 W/mK (EN13165)]. TGA analysis revealed that the composite had a good thermal stability, while the FT-IR analysis showed that the phenyl groups of polystyrene decreased the H-bonding of the composite. SEM analyses show that the blowing agent decreased the homogeneity of the composite. The findings suggest that buffing dust can be re-used to prepare a thermal insulation when being mixed with polystyrene in the presence of a blowing agent. This would enhance the insulation properties of the material without compromising its mechanical properties. Although the manufacturing of the insulation material is widely known, how to further process the material as a marketable product remains the bottleneck of this work. Overall, this composite, resulting from the mixture of buffing dust and polystyrene, represents one of the most promising options as a thermal insulation material in the construction industry.

**Author Contributions:** Conceptualization, A.Q.A., M.H.D.O.; methodology, M.J.; validation, S.N.; Analysis, S.H.; investigation, B.M.; resources, W.U.; writing—original draft preparation, A.M.; writing—review and editing, T.A.K.; supervision, A.M.; project administration, M.A. (Muhammad Amjad).; Visualization, A.A., M.A. (Muhammad Arif). All authors have read and agreed to the published version of the manuscript.

**Funding:** This research received no external funding.

**Institutional Review Board Statement:** Not applicable.

**Informed Consent Statement:** Not applicable.

**Data Availability Statement:** Not applicable.

**Conflicts of Interest:** The authors declare no conflict of interest.

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
