# Peer review of "Reuse of Buffing Dust-Laden Tanning Waste Hybridized with Poly- Styrene for Fabrication of Thermal Insulation Materials"

_sustainability, doi:10.3390/su15031958_

Round 1
Reviewer 1 Report
Thank you for submitting your manuscript. This research focuses on the reuse of waste (i.e., buffing dust) produced from tanning industries as thermal insulation materials for construction. The aim is to produce efficient insulation materials to reduce building energy consumption, a critical area for environmental sustainability. Notably, this is an important area of research, nonetheless, critical gaps remain in the manuscript and needs to be addressed for publication. Please see my comments below:
- The abstract of the manuscript does not clarify why the reuse of buffing dust (from leather tanning waste) matters. The reviewer would suggest providing more context to the reader on why this reuse matters and what is the environmental impact that this waste currently have. The first sentences of the introduction provide good context to the reader, such sentences (or a variation of those) would be useful on the abstract.
- The authors mention Circular Economy in lines 48-50, nonetheless, no context is provided to the reader on what is a Circular Economy. I suggest providing such context right after this paragraph, examples include what is a Circular Economy, why does it matter, what is the contrast to the current linear economic model.
- The authors mention production of “thermal insulation materials” with buffing dust…is there a particular material that the authors are focusing? e.g., acoustical ceiling panels – ACT.
- Line 90: “Although many people have intensified their work to find out the process to re-use the buffing waste [41-43], to the best of our knowledge, buffing dust has not been mixed with polystyrene to form composites in the presence of blowing agent.” – The authors could further explain why mix buffing waste with polystyrene is important, what specifically is the benefit of the methodology proposed as opposed to the other authors that you pointed [41-43].
- Section 5 about the implications of such work to the Circular Economy is very superficial and needs to be flushed out better. For example, the authors could step back and discuss what it would take for the construction industry to actually implement buffing dust in acoustical ceiling panels production (e.g., what it would take to the fabricators of this material?). Additionally, the authors could discuss more about the logistics of such implementation; for example, is it actually feasible in countries with minimal leather tanning industry? Are the countries with significant leather tanning industries the same producing acoustical ceiling panels?
Review the entire manuscript looking for grammar mistakes and misspellings. For example: “scientists has worked” (Line 48); “Implications of this work on circlar economy” (Line 257); etc.

Author Response
Manuscript ID: Sustainability-2107878
Title: Reuse of buffing dust-laden tanning waste hybridized with polystyrene for fabrication of thermal insulation materials
The authors are thankful to the Reviewers for their constructive feedbacks, which have improved the quality of the work before its publications.
Reviewer 1
- Comment
The abstract of the manuscript does not clarify why reuse of buffing dust (from leather tanning waste) matters. The reviewer would suggest providing more context to the reader on why this reuse matters and what is the environmental impact that this waste currently have.
Reply to Comment:
“Overall, this work not only resolved the energy problems, but it also brought positive impacts on the environment by recycling hazardous buffing dust and then converting and reusing it as a thermal insulation material. Not only this reduces air pollution, resulting from the buffing dust, but this also promotes its resource recovery in the framework of circular economy (CE).”
As suggested, the explanation why the reuse of buffing dust matter and and what is the environmental impact that this waste currently has have been incorporated in the text. Please refer to the changes on page 1 line 30-32.
- Comment
The first sentences of the introduction provide good context to the reader, such sentences (or a variation of those) would be useful on the abstract.
Reply to Comment:
“Due to rapid industrialization, air pollution, resulting from buffing dust waste produced by leather tanning industry, has become a critical issue for the environment and public health.”
As suggested, the first sentence of the introduction has been revised and modified accordingly for providing good context in the revised abstract on page 1 lines 16-17.
- Comment
The authors mentioned Circular Economy (CE) in lines 48-50. Nonetheless, no context was provided to the reader on what is a CE. I suggest providing such context right after this paragraph, examples include what is a Circular Economy, why does it matter, what is the contrast to the current linear economic model.
Reply to Comment:
…, while exploring ways for recycling and then reuse it in a circular economy (CE) after incorporating technological values. For the sake of sustainability, the waste needs to be reduced, recovered, recycled, and reused in such a way that it could be turned into a valuable resource such as thermal insulation material to promote resource recovery. Resource recovery minimizes waste generation by utilizing by-products that remain from other processes, adopting a zero-waste approach.
As compared to linear economy, which results in environmental challenges such as CO2 emission, a circular economy facilitates waste prevention/reduction and its recovery and recycling to deal with the impacts of waste disposal on the nexus of society, environment, and economy. The circular economy leads to a way of reverting excessive rates of waste generation and resource consumption, while failing linear economic models treat resources as if they are infinite. Unlike CE, linear economy is unsustainable.
As suggested, a brief overview on circular economy (CE) has been incorporated in the revised text on page 2 lines 57-68.
- Comment
The authors mention production of “thermal insulation materials” with buffing dust… Is there a particular material that the authors are focusing? e.g., acoustical ceiling panels – ACT.
Reply to Comment:
Conventional acoustical ceiling panels (ACT) do not stop heat exchange effectively in buildings. However, the composite that forms “thermal insulation material” stops the heat transfer. Therefore, it was tested as thermal insulation material and reused as ceiling panels.
Please refer to the change on pages 3 lines 104-105.
- Comment
Line 90: “Although many people have intensified their work to find out the process to re-use the buffing waste [41-43], to the best of our knowledge, buffing dust has not been mixed with polystyrene to form composites in the presence of blowing agent.” – The authors could further explain why mix buffing waste with polystyrene is important, what specifically is the benefit of the methodology proposed as opposed to the other authors that you pointed [41-43].
Reply to Comment:
To enable the composite to have low thermal conductivity and to increase compressive strength, the buffing dust was mixed with polystyrene. Installing it in buildings as thermal insulators is important because the materials could reduce energy consumption, while allowing the recovery of tanning waste and the recycling of unused polishing powder to cope with hazardous air pollution, resulting from the leather industry.
Please refer to the changes on page 3 lines 120-125.
- Comment
Section 5 about the implications of such work to the Circular Economy is superficial and needs to be flushed out better. For example, the authors could step back and discuss what it would take for the construction industry to actually implement buffing dust in acoustical ceiling panels production (e.g., what it would take to the fabricators of this material?).
Reply to Comment:
To reflect the ways of “thinking globally, acting locally” in promoting sustainability, for the sake of waste avoidance, industrial users strive for material recovery from downstream processing to address air pollution problem due to buffing dust from tanning industry [51]. This work has uncovered practical applications for synthesizing thermal insulation panels that can be used for the ceiling of buildings in the construction industry [52].
The recovery, recycle, reuse, conversion and re-introduction of the buffing dust waste into a supply chain of ceiling panels for the production of insulation materials contribute to a sustainable production and consumption in the construction industry with respect to the circular economy [53]. If the growth of the waste could be reversed by tackling the buffing dust from upstream to downstream, this facilitates a sustainable resource recovery from by-products in tanning industry [54].
This work represents a practical solution for air pollution problems due to buffing dust waste because the waste, recovered from downstream process, can be recycled and reused to produce materials with added value after being mixed with polystyrene to improve its compressive strength [55]. This represents a closed-loop of circular economy that ultimately benefits the construction industry [56]. Shifting from a linear economy to circular economy is vital to manage buffing dust efficiently and effectively [57].
The insulation panels bring socio-economic benefits that can minimize energy consumption [52]. This reduces the tanning waste disposed of in landfills [58]. This paradigm enables environmental problems due to waste generation to be tackled locally to bring tangible impacts on the environment in the long-term [59]. In the future, additive materials and new methods can be tested to enhance the physico-chemical and mechanical properties of the waste such as water absorption, density, mechanical strength, and thermal conductivity [60].
Please refer to the change on page 8 lines 272-292.
- Comment
Additionally, the authors could discuss about the logistics of such implementation. For example, is it feasible in countries with minimal leather tanning industry? Do the countries with significant leather tanning industries have the same producing acoustical ceiling panels?
Reply to Comment:
With respect to the logistics of such implementation, it is not economically feasible in areas with minimum leather tanning industry. To reverse the growth of the buffing dust from upstream to downstream and to facilitate a sustainable resource recovery from by-products in tanning industry, extensive presence of leather tanning industry is necessary to produce ceiling panels.
Please refer to the changes on page 9 lines 282-286.

Reviewer 2 Report
Dear Authors, please following these recommedations.
1) Please, correct the first sentence in introductory part (line 34).
2) Please, reformat figure 1.
3) Please, reformat figure 6.
4) Please, correct line 257 (remove circlar and insert circular).
5) Please, insert in Conclusion paragrah a short description of limitations of this study.

Author Response
Manuscript ID: Sustainability-2107878
Title: Reuse of buffing dust-laden tanning waste hybridized with polystyrene for fabrication of thermal insulation materials
The authors are thankful to the Reviewers for their constructive feedbacks, which have improved the quality of the work before its publications.
Reviewer 2
- Comment
Please correct the first sentence in introductory part (line 34).
Reply to Comment:
As the second largest industry in Pakistan, tanning industry grows day after day due to economic growth. About 800 units of tanneries are located in Kasur (Pakistan). They produce a large amount of buffing dust waste that contribute to air pollution in the city. As a result, recently, environmental pollution has increased at alarming point due to rapid industrialization [1].
As suggested, the first few sentences of the introduction section have been revised accordingly. Please refer to the changes on page 1 lines 41-44.
- Comment
Please reformat Figure 1
Reply to Comment:
Figure 1 has been reformatted in the revised manuscript on page 4.
- Comment
Please reformat figure 6.
Reply to Comment:
Figure 6 has been reformatted in the revised manuscript on page 7.
- Comment
Please correct line 257 (remove ‘circlar’ and insert ‘circular’).
Reply to Comment:
The word “circular” has been corrected on page line 8 line 271.
- Comment
Please insert in Conclusion section a short description of limitations of this study.
Reply to Comment:
Although the manufacturing of the insulation material is widely known, how to further process the material as a marketable product remains the bottleneck of this work.
As requested, the limitation of this wrk has been incorporated in the revised text on page 9 lines 318-319.

Round 2
Reviewer 1 Report
Thank you for your revision. Per my review, there are a few key points yet to be addressed before publication:
- The first sentence in the abstract is not very clear. I would revise it as follow “Air pollution resulting from buffing dust waste produced by the leather tanning industry has become a critical issue for the environment and public health.”
- The last “(CE)” in the abstract is not useful, please remove it.
- No need to separate the first paragraph from the second in the Introduction. Please keep it as a single paragraph.
- While the authors added a bit more context about sustainability in the introduction, the manuscript still does not incorporate a single definition of Circular Economy, which is unacceptable given that the authors repeatedly claim that incorporating buffing dust would promote circularity in the construction industry.
- Line 104 - The authors describe that acoustic ceiling panels have been tested as insulation materials but could not stop head exchange effectively. Please include a reference that supports this.
Author Response
Manuscript ID: Sustainability-2107878
Title: Reuse of buffing dust-laden tanning waste hybridized with polystyrene for fabrication of thermal insulation materials
The authors are thankful to the Reviewer 1 for his/her constructive feedbacks, which have improved the quality of the work before its publications.
Reviewer 1
- Comment
The first sentence in the abstract is not very clear. I would revise it as follow “Air pollution resulting from buffing dust waste produced by the leather tanning industry has become a critical issue for the environment and public health.
Reply to Comment:
Air pollution resulting from buffing dust waste produced by the leather tanning industry has become a critical issue for the environment and public health.
As suggested, the first sentence of the abstract has been revised. Please refer to the changes on page 1 line 17-18.
- Comment
The last “(CE)” in the abstract is not useful, please remove it.
Reply to Comment:
…., but this also promotes resource recovery in the framework of a circular economy.”
As suggested, the last “(CE)” has been removed in the revised abstract on page 1 line 39.
- Comment
No need to separate the first paragraph from the second in the Introduction. Please keep it as a single paragraph.
Reply to Comment:
As the second largest industry in Pakistan, tanning industry grows day after day due to economic growth and industrialization. About 800 units of tanneries are located in Kasur (Pakistan). They produce a large amount of buffing dust-laden tanning waste that contribute to air pollution in the city. As a result, recently, air pollution has increased at alarming point due to the waste [1]. As a hazardous waste, the buffing dust contains Cr(III). Under certain conditions, it is converted into Cr(VI). Traditional waste disposal methods such as landfill and incineration could not remove the hazardous nature of the toxic metal to the environment [2-4]. Consequently, the tanning waste causes adverse effects on public health [5-6].
As suggested, the paragraph has been consolidated in the revised text on pages 1-2 lines 42-50.
- Comment
While the authors added a bit more context about sustainability in the introduction, the manuscript still does not incorporate a single definition of Circular Economy, which is unacceptable given that the authors repeatedly claim that incorporating buffing dust would promote circularity in the construction industry.
Reply to Comment:
As suggested, the definition of circular economy has been incorporated in the revised text on page lines.
Circular economy is defined as ‘an economic system based on the reuse and regeneration of materials or products, especially as a means of continuing production in a sustainable or environmentally friendly way.’
Please refer to the change on page 2 lines 67-69.
- Comment
Line 104 - The authors describe that acoustic ceiling panels have been tested as insulation materials but could not stop heat exchange effectively. Please include a reference that supports this statement.
Reply to Comment:
As suggested, the following reference has been cited to support the statement on page 3 line 109:
[40] Dirisu, J.O.; Fayomi, O.S.I.; Oyedepo, S.O. Thermal emission and heat transfer characterictis of ceiling materials: A necessity. Energy Proc. 2019, 157, 331-342.
Please refer to the changes on page 11 lines 410-411.
